# The Role of BDNF, YBX1, CENPF, ZSCAN4, TEAD4, GLIS1 and USF1 in the Activation of the Embryonic Genome in Bovine Embryos

**DOI:** 10.3390/ijms242216019

**Published:** 2023-11-07

**Authors:** Bingnan Liu, Jiaxin Yan, Junjie Li, Wei Xia

**Affiliations:** 1College of Animal Science and Technology, Hebei Agricultural University, Baoding 071000, China; liubingnan139@139.com (B.L.); qianyelinxi@outlook.com (J.Y.); lijunjie816@163.com (J.L.); 2Research Center of Cattle and Sheep Embryo Engineering Technique of Hebei Province, Baoding 071000, China

**Keywords:** bovine, zygotes/embryos, genetic program, zygotic transcription

## Abstract

Early embryonic development relies on the maternal RNAs and newly synthesized proteins during oogenesis. Zygotic transcription is an important event occurring at a specific time after fertilization. If no zygotic transcription occurs, the embryo will die because it is unable to meet the needs of the embryo and continue to grow. During the early stages of embryonic development, the correct transcription, translation, and expression of genes play a crucial role in blastocyst formation and differentiation of cell lineage species formation among mammalian species, and any variation may lead to developmental defects, arrest, or even death. Abnormal expression of some genes may lead to failure of the embryonic zygote genome before activation, such as BDNF and YBX1; Decreased expression of CENPF, ZSCAN4, TEAD4, GLIS1, and USF1 genes can lead to embryonic development failure. This article reviews the results of studies on the timing and mechanism of gene expression of these genes in bovine fertilized eggs/embryos.

## 1. Introduction

The prerequisites for pregnancy require oocyte maturation, successful fertilization, and the acquisition of a high-quality embryo. Most infertility in both humans and animals is caused by alterations in various developmental regulatory factors, such as mutations in multiple developmental regulators. These causes may result in impaired oocyte maturation, failure of fertilization, or cessation of embryo development at an early stage. Spermatogonia undergo proliferation, growth, and maturation stages to develop into mature spermatozoa, but to achieve fertilization; they also need to undergo the “acrosome reaction” and the energization stage, while the development of the primordial oocyte into a “competent” egg cell is even more difficult. The maturation of the oocyte is not continuous; after undergoing the proliferative process of mitosis, it undergoes meiosis, but this does not allow it to develop to maturity, and after undergoing the first meiosis, the follicle ruptures and is expelled. At this point, the oocyte is not capable of fertilization and needs to undergo a second meiosis to develop into a capable fertilization oocyte. After fertilization in mammals, the fertilized egg undergoes cleavage and division to develop into a blastocyst. During this period, there were three important developmental events: chromosome group activation, zygotic genome activation (ZGA), polarization/compression, and the first specification of the cell lineage [1,2]. Two distinct cell types emerge, the inner cell mass (ICM) and the trophoblastic ectoderm (TE), which subsequently form the blastocyst and then continue to differentiate. Most animals, including humans, are influenced by maternal genes prior to ZGA, and embryonic development changes from maternal factor-dependent to embryonic factor-dependent after ZGA. However, all processes involved in cell differentiation, as well as in a range of embryonic development, are closely linked to changes in gene expression and histones. In mammals, the correct transcription, translation, and expression of genes play a pivotal role in the formation of blastocysts and the differentiation of cell lineage species. During the development of bovine embryos, different genes are expressed at different stages, and the expression of these genes is regulated by many factors, including transcription factors and epigenetic modifications. The study of these genes and their regulatory mechanisms will lead to a better understanding of the genetic program of bovine embryonic development. Many maternal genes are synthesized and accumulated in the oocyte and play a key role in early embryonic development. Among them, the maternal effector genes (MEGs) refer to the genes that play an important role in maintaining the survival and development of mammalian embryos during the cleavage stage after fertilization. For example, knocking out NOBOX in bovine embryos significantly reduced not only the blastocyst rate but also the expression of pluripotent genes (POU5F1/OCT4 and NANOG) and the number of cells in the inner cell mass within the blastocyst [3]. ZNFO has intrinsic transcriptional repressor activity and is another maternally derived oocyte-specific nuclear factor essential for early bovine embryonic development. Knockdown of ZNFO by siRNA significantly reduced embryo development to the 8–16 cell stage and blastocyst stage [4]. Maternal genes can support early embryonic development before the syn gene is activated and then begins to degrade. Previous studies have found that nearly 90% of maternal genes are degraded when cattle congenital gene activation occurs at the 8–16 cell stage [5]. Since the embryonic genome is transcribed up to the ZGA stage, it is likely that the accumulation of maternal factors (proteins and mRNAs) during oocyte development has led to the recording of this phenomenon. In addition, the requirement for extensive epigenetic reorganization prior to embryo implantation is associated with the pluripotency and activation of the embryonic genome. Transcription of the zygotic genome is required for the normal development and differentiation of early animal embryos, and ZGA-related factors are essential for the transcriptional regulation of the zygotic genome. For instance, OCT-4, a transcription factor that plays a pivotal role in maintaining the pluripotency of embryonic stem cells (ESCs), has been demonstrated to elicit a diminished rate of blastocyst development in bovine embryos with targeted knockdown of OCT-4 via siRNA injection [6,7]. It has been confirmed that the knockdown of SOX2, an additional transcription factor associated with embryo quality and essential for maintaining embryonic pluripotency, leads to a reduction in the number of blastocysts and blastomeres in cattle [8,9,10]. Knockdown of TSPY by microinjection in bovine fertilized eggs had no effect on the female embryo population but caused male embryo development to stop before the blastocyst stage [11].

In recent years, an increasing number of genes associated with oocyte maturation and early embryonic development have been identified. In order to successfully develop into healthy offspring, growing and maturing mammalian oocytes and embryos must undergo matching gene expression and epistatic modifications before they can develop into new life. Therefore, knowledge of key genes and epistatic modifications for normal development before implantation is essential to ensure normal gametogenesis, normal fertilization and maintenance of high fecundity rates, and normal early embryogenesis in domestic animals. Compared with natural mating and artificial insemination, the success rate of in vitro production (IVP) of bovine embryos to establish pregnancy is low, and oocyte quality is a key factor limiting the success of animal pregnancy. Therefore, in order to better understand the genetic regulation of early embryo development and improve the efficiency of in vitro embryo production, we summarized the genetic factors that have been found to inhibit oocyte maturation in recent years.

## 2. Adverse Effects of Abnormal Gene Expression on ZGA

### 2.1. BDNF (Brain-Derived Neurotrophic Factor) Is Associated with Egg Maturation

BDNF is detected in numerous mammals, including ovaries and oocytes, and holds significance in the progression of mammalian follicles, ovulation, proliferation of granulosa cells, fertilization, and subsequent embryonic development [12,13,14,15,16]. A variety of animals (such as rodents, sheep, cattle, etc.) can secrete high-affinity neurotrophins (NTs) in the ovaries, and there are p75NTR receptors, which play an important role in signal transduction. High-affinity receptors for neurotrophin-4/5 (NT-4/5) and BDNF are important in early follicular growth and oocyte survival, and BDNF is secreted by cumulus and granulosa cells, so it is speculated that they can affect oocyte maturation and early embryonic development in many species [17,18,19].

Zhao et al. first used immunofluorescence staining to detect the expression pattern of BDNF and its receptors during early buffalo embryonic development and found that BDNF expression first increased and peaked at the 4-cell stage and then gradually decreased (Figure 1) [20]. The mRNA manifestation profile of BDNF was akin to that of its receptor NTRK2, and the manifestation of an alternative receptor, p75, was significantly limited in comparison to NTRK2. The high synchronization of the two suggests that both are involved in follicular development. When buffalo oocytes were treated with K-252a and p75 inhibitor (pep5) for in vitro maturation, the reaction rate was significantly reduced when BDNF and receptor inhibitor K-252a were added simultaneously, demonstrating that K-252a can eliminate the effect of BDNF on oocyte maturation, receptor NTRK2 works with BDNF during maturation of buffalo oocytes The receptor NTRK2 acts with BDNF during oocyte maturation, while the receptor p75 does not seem to play a role in oocyte maturation. Therefore, it is concluded that p75, as a low-factor receptor, may not function primarily in buffalo granulosa cells.

The high synchronization of BDNF with its receptor NTRK2 suggests that both are involved in follicle development, so it is speculated that the enhancement of the developmental capacity of buffalo embryos may be related to BDNF. A discovery was made that the utilization of 10 ng/mL BDNF greatly enhanced the rate of oocyte maturation and blastocyst formation in buffalo embryos using in vitro fertilization. Nevertheless, as the concentration increased to 100 ng/mL, there was a decline in the blastocyst rate, indicating a two-way effect of BDNF [20]. To further study the mechanism of BDNF on cumulus cells, the mRNA expression changes of apoptosis-related genes and BDNF receptor genes during IVM and the selection of developmental genes in cumulus cells were analyzed by RT-qPCR. Further study on the mechanism of BDNF’s action on cumulus cells during IVM finally found that BDNF can down-regulate apoptosis-related genes CASP9 and FAS. The expression levels of receptor-related genes NTRK2 and cumulus cell development-related genes CCNB1, PCNA, GJA4, GJA1, HAS2, PTX3, and TNFAIP6 were up-regulated, which enhanced the proliferation of cumulus cells, promoted the expansion of cumulus cells, and thus improved the maturation rate of buffalo oocytes. It can also promote the proliferation of cumulus cells.

### 2.2. Y-Box Binding Protein 1 (YBX1) Reduces Attenuation of Damaging Maternal Genes

YBX1 has a significant function in the stabilization of RNA and regulation of transcription. This particular gene encodes a protein with a cold shock domain that is highly conserved and possesses broad binding properties toward nucleic acids. The encoded protein serves as a multifaceted nucleic acid-binding macromolecule, demonstrating its involvement in manifold cellular endeavors encompassing the orchestration of transcription and translation, intricate pre-mRNA splicing, intricate DNA reparation, and delicate mRNA packaging. Additionally, this protein is an integral constituent within messenger ribonucleoprotein (mRNP) formations, suggesting potential implications in the intricate realm of microRNA processing [21]; Y-Box Binding Protein is reported to be enriched in oocytes [22,23] and recognized as primary constituents of cytoplasmic messenger ribonucleoproteins (mRNPs) with a diverse array of RNA-binding capacities [24]. Deng et al. used the public RNA-seq dataset to reanalyze the transcription level of YBX1 in bovine embryos. Bovine YBX1 was progressively up-regulated from oocyte to blastula, and during the early development of bovine embryos, YBX1 expression was significantly increased in 8-cell embryos [25] (Figure 1). YBX1 is highly expressed in bovine mature oocytes, and its expression is further increased after fertilization, especially during MZT, as confirmed by Grurgia et al. [26].

In order to further study the effect of YBX1 on embryonic development ability, siRNA injection was performed to knock out the expression of YBX1 and observe the expression changes of YBX1. A total of 5154 differentially expressed genes (DEGs) were obtained by using DESeq2. It was found that in the embryos with low YBX1 knockdown, the number of 2-cell and 4-cell stage blocked embryos increased, and the percentage of blastocysts decreased significantly. Among them, 1623 and 3531 up-regulated and down-regulated genes were enriched in the regions that regulate RNA splicing and RNA stability. Moreover, a large number of genes related to Z-decline were changed, and these results all indicated that YBX1 knockdown would lead to the impairment of alternative splicing (AS) and RNA stability during ZGA, as well as the attenuation of maternal mRNA [25].
Figure 1Description of expression patterns of BDNF and YBX1mRNA. The horizontal coordinate represents the stages of embryonic development; The ordinate only represents the rise and fall; there is no actual value.
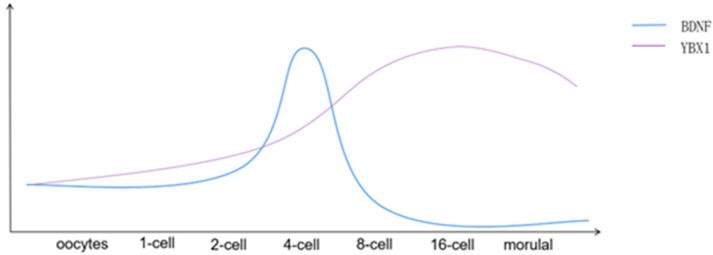


## 3. Adverse Effects of Abnormal Gene Expression on Zygotic Genome Activation

### 3.1. Down-Regulation Centromeric Protein F (CENPF) May Cause the Embryo to Stagnate at the 8-Cell Stage Formatting of Mathematical Components

CENPF is a component of the centromere-centromeric complex, and the expression of this gene results in the production of a protein that is linked to this complex. This protein plays a critical role in the differentiation of somatic cells. It was found that the number of CENPF mRNA decreased gradually from 2 to 8 cells stage, and all of them originated from the mother line. The transcription of CENPF in embryo begins at the late stage of 8 cells, so it can be assumed that after the appearance of EGA, its gene expression begins to rise again and remains basically unchanged until the blastocyst stage (Figure 2) [27]. To investigate the impact of CENPF on embryonic development potential, Tereza et al. employed CENPF-specific double-stranded RNA (dsRNA) to suppress the corresponding mRNA expression and found that the embryos showed obvious developmental abnormalities until the EGA stage and found that the most common defects in the embryos inoculated with CENPF dsRNA were: The size varies, the edge of the blastomere is blurred, the part of the blastomere becomes transparent, there are obvious nuclear fragments in the embryo, even there is no nucleus in the blastomere, only less than 1/3 of the embryos developed to 8 cells [28]. CENPF plays an important role in the aggregation, arrangement, and separation of chromosomes because CENPF plays an important role in the interaction between centromere and microtubule in somatic cells [29]. The protein functions as a vital component of the nuclear framework during the G2 stage of interphase. As G2 nears its conclusion, the protein establishes connections with the kinetochore and maintains this interaction until early anaphase. It is found in the spindle midzone during late anaphase and the intracellular bridge during telophase and is thought to undergo subsequent degradation. The precise pattern of localization for this protein suggests its potential role in promoting chromosome segregation during mitosis. Studies in humans and other animals have shown that CENPF silencing affects cell division by disrupting the chromosome division process [30,31]. The research results of cattle also showed that blastomere abnormalities, to a certain extent, can also be understood as related to the process of chromosome division, but the specific mechanism or pathway through which the impact on bovine embryos remains to be explored.

### 3.2. Zinc Finger and SCAN Domain Containing 4 (ZSCAN4) May Result in 16-Cell Phase Growth Arrest

ZSCAN4 is an extraordinary gene responsible for encoding a protein that participates in the upkeep of telomeres. This remarkable gene plays a pivotal role in a crucial attribute of mouse ESs, namely, defying cellular senescence and maintaining normal karyotypes for many cell divisions in culture. Initially, its presence was discovered to be exclusive to the late 2-cell stage of the pre-implantation embryo in mice [32,33]. Knockout of ZSCAN4 with interfering RNA(siRNA) results in delayed progression at the 2–4 cell stages, leading to the failure of embryo implantation [32]. However, the expression status and role of ZSCAN4 in the pre-implantation development of bovine embryos remains unclear. Kazuki TAKAHASHI et al. investigated the necessity of the development of the ZSCAN4 gene before implantation in bovine embryos. They found that the expression of ZSCAN4 remained in a low pattern until the 4th cell stage but increased after the 4th cell stage, where the expression level was significantly increased at the 4–8 cell stage until the embryonic transcription level peaked at the 16th cell stage (Figure 2). In addition, it was found that ZSCAN4 increased significantly at the 4–8 cell stage due to de novo synthesis of ZSCAN4 in zygotes [34]. Additionally, they cultured bovine embryos that were injected with ZSCAN4-siRNA in vitro. The results demonstrated that all embryos encountered growth arrest at the 16-cell stage, with only a limited number progressing to the blastocyst stage. Moreover, the researchers assessed the mRNA levels of developmental pluripotency-associated gene 2 (DPPA2) and Piwi-like RNA-mediated gene silencing 2 (PIWIL2) in 16-cell embryos to evaluate the impact of reducing ZSCAN4 expression on reprogramming-associated gene transcripts. It was found that PIWIL2 expression levels were reduced in embryos injected with ZSCAN4-siRNA. Research has indicated that the Piwi protein and its linked small RNAs, known as Piwi interacting RNAs (piRNAs), impede the transcription of late transposition factors in animal germ cells, resulting in a notable upregulation of long terminal repeat retrotransposons [35,36], among them, pi-RNAs have been shown to be essential for targeted elimination of mRNA transcripts during the [37]. In addition, more biological reactions occur during the transcription of ZSCAN4, including instantaneous expression of other ZGA-specific groups, rapid expansion of telomere 5, and blocking of translation of the entire protein [38,39]. It can be speculated that the down-regulation of PIWIL2 expression will cause the dysfunction of other retrotransposons (including transcriptional transposons) in bovine embryos, and thus halt the early development of ZGA. However, the mechanism of the interaction between ZSCAN4 and PIWI-piRNA in bovine embryos is unclear and remains to be investigated.

### 3.3. Interaction of TEA Domain Transcription Factor 4 (TEAD4) and CCN2

During embryonic development, TEAD4 plays an important role in organ formation and development by regulating gene transcription. Studies have shown that TEAD4 deficiency leads to early embryo death and developmental deformities [40,41]. TEAD4 is a regulator of blastomere TE properties in mouse models and plays a key role in TE differentiation [40,42], participating in a variety of life processes such as cell proliferation, cell survival, tissue regeneration, and stem cell maintenance; TEAD4 potentially plays a role in the development of porcine embryo blastocysts by modulating the activity of SOX2, thereby influencing the conversion of morula into blastocyst [43]; bovine TEAD4 has a unique function to activate the specific pregnancy recognition factor interferon tau in ruminants [44]. More studies have shown that TE is a single layer of epithelioid cells surrounding the outer layer of the blastula, and the expression level of transcription factor caudal homo box 2(CDX2) determines its development direction. During embryonic development, CCN2 is involved in the regulation of many growth factors and extracellular matrix interactions and has different expression patterns and effects in different organs and tissues [42,45]. Therefore, CDX2 and CCN2 play important roles in blastocyst development. In addition, CCN family 2(CCN2) is an important downstream gene of TEAD4, which is widely used in mammalian somatic cells, and TEAD4 regulates the proliferation of CCN2 by regulating its expression. Bovine TEAD4 was studied by Hiroki Akizawa et al. It was found that TEAD4 mRNA was expressed in TE and ICM, and Tead4 mRNA content was higher in TE than in ICM, and Tead4 was mainly expressed in TE nucleus. The initial amount of TEAD4mRNA was low and increased from the 8-cell stage, and TEAD4 reached its peak after the morula stage (Figure 2). By using short hairpin RNA (shRNA) to interfere with the TEAD4 gene, TEAD4 was knocked out KD, and it was found that CDX2, GATA2, and CCN2 genes were significantly reduced, and the change of CCN2 expression was the most prominent [46]. When CCN2KD was performed in bovine embryos, TEAD4 expression levels were significantly reduced in CCN2KD blastocysts. It is worth noting that the expression level of TEAD4 in CCN2KD blastula also showed a significantly decreased trend. Interestingly, neither TEAD4KD nor CCN2KD had an effect on cleavage development rate or blastocyst formation in vitro, but CCN2KD led to a significant reduction in the ratio of TE to ICM cell numbers without changing the total number of TE and ICM cells. Regulation of cell composition in bovine blastula is related to the expression of CCN2 in TE; since bovine CCN2 is expressed in endometrial epithelial cells, it is speculated that maternal CCN2 may influence pre-implantation development. These results indicate that TEAD4 and CCN2 regulate and influence each other, and TEAD4 directly regulates the transcriptional activation of CCN2, leading to changes in cell characteristics [47,48]. The expression of CCN2 is also affected by TEAD4. The interaction between TEAD4 and CCN2 is important for normal cell differentiation during pre-implantation development.

### 3.4. Deletion of GLI-Similar 1 (GLIS1) May Lead to ZGA Failure

GLIS1 is a transcription factor 15 residue closely related to the Gli family [49,50,51]. GLIS1 plays an important role in the formation and development of organs such as the cardiovascular system, kidney, eye, thyroid, and pancreas and is considered to be a direct recombinant coding factor that can promote the production of pluripotent stem cells and is richly expressed in both unfertilized mouse oocytes and 1-cell stage embryos [52,53]. In addition, Glis1 also has the function of temporal and spatial regulation, so it can be speculated that Glis1 regulates the embryonic development process in a certain period of time [50,54]. Kazuki Takahashi et al. studied GLIS1 in bovine oocytes in vitro and found that a large amount of the GLIS1 gene could be detected in both bovine oocytes and embryos at stage 1 to 4 cells, and the GLIS1 gene decreased from the stage 1 cell to the stage of 8 cells and beyond (Figure 2) [55]. By injecting Glis1-siRNA into bovine embryos to investigate the relationship between the effects of bovine embryo development and the downregulation of the GLIS1 gene, it was found that the injected embryos had no effect on 16-cell stage development, but the rate of 32-cell stage development was significantly reduced. In order to further explore the effect of GLIS1 downregulation on gene transcripts, mRNA expressions of PGK1, PDHA1, heat shock homologous protein 70 (HSPA8), and X non-live specific transcripts (XIST-) at cell stage 8–16 were detected. The expression of PDHA1 and HSPA8 decreased significantly. PDHA1 is involved in glucose metabolism and plays a key role in embryonic development, especially in cattle and mice [56,57]. HSPA8 encodes HSC70, which is involved in the pretreatment and selective autophagy of intron RNA, so HSC70 knockdown will lead to a large number of cell death of various types. In conclusion, GLIS1 down-regulated embryos may lead to the failure of ZGA initiation, suggesting that GLIS1 may be an important factor in the pre-implantation development of bovine embryos.

### 3.5. Upstream Stimulating Factor 1 (USF1) Gene Knockout Affects Early Embryonic Development in Cattle

USF1 is a transcription factor with a basic helix-loop-helix structure that selectively attaches to E-box DNA motifs. It is recognized as a cis-element of crucial genes responsible for oocyte expression, which is vital for early embryonic and oocyte development [58]. Datta T et al., therefore, first examined the expression patterns of USF1 in bovine oocytes and embryos at different times. USF1 mRNA was found to increase during meiosis, increase significantly during 2–8 cells, and then decrease until it is almost undetectable at the blastocyst stage, indicating that it may play a role in embryo genome activation, indicating that the gene is maternal in origin and may be consumed or degraded during embryo genome activation (Figure 2). In order to investigate the role of USF1 in bovine oocyte and embryonic development, the siRNA program was used to mediate gene silencing in bovine embryos, and the abundance of USF1 transcripts was significantly reduced after injection of USF1 siRNA. The study found that the total cleavage rate in bovine embryos after USF1 knockout had no effect but reduced the number of development to the 8–16 cell stage, and in particular, the blastocyst rate was significantly reduced. In addition, these genes TWIST 2, JY-1, GDF 9, and FST were found to carry USF1-binding elements (e-boxes) in their promoter region necessary for the ability of bovine oocytes to develop. The abundance of TWIST2 and JY1 mRNA increased, but the abundance of GDF9 and FST transcripts decreased in USF1 siRNA oocytes collected at the MII stage. The abundance of GDF9 transcripts was moderately decreased, suggesting that negative control siRNA had a moderate off-target effect on GDF9 expression. The transcription factor TWIST2 functions as a molecular switch, which can either activate or suppress target genes by directly binding to conserved E-box sequences in promoter regions and enlisting coactivators or suppressors [59]. As such, USF1 has the potential to regulate the transcriptional levels of GDF9, FST, TWIST2, and JY-1 during oocyte maturation.

## 4. Conclusions

After fertilization, mammals may require a specific object or abundant eggs for mRNA transcription and protein synthesis to give them the ability to develop fully. In mice, blastocyst formation is dependent on the presence of maternal factor(s) as mRNA in the egg and also on syncytial genetic information [60,61]. The normal expression of these genes is inextricably linked to the normal development of the blastocyst and the normal attachment of the early embryo. Moreover, abnormal oocyte mRNA level abundance and pattern of oocyte–follicle axis of development may lead to oocyte development failure and thus affect later development [62]. Large mammals similar to mice, such as cattle and sheep, have similar mechanisms. Among them, transcripts of genes required for oocytes and zygotes of some domestic animals have been identified, and different levels of transcript deletion cause developmental arrest and other developmental disorders at different stages, including BDNF, GLS1, YBX1, GENPF, ZSCAN4, and TEAD4, and at which stage they play a key role, as shown in Figure 3. A general pattern emerges from various studies: During the continuous division and maturation of the primordial follicle, a series of transcriptome changes occur. Two attenuation occurs during the period; the first is M-attenuation, starting from the GVBD stage. The second attenuation from the MII phase is called Z-attenuation; after fertilization, maternal mRNA will be heavily degraded via a key developmental process known as maternal to zygotic transition (MZT), in which developmental control is transferred from maternally supplied gene products to products synthesized from the zygotic genome, resulting in activation of the syngeneic genome [63,64,65,66]. Thus, oocyte-derived mRNAs and proteins may play an important role in this process [67].

For example, if BDNF is defective during the maturation and early embryonic development of buffalo follicles and oocytes, the expression levels of related genes and receptor genes are dysregulated so that cumulus cells and their receptor NTRK2 cannot promote oocyte maturation. However, after YBX1 gene knockdown, it affects the stability of AS and RNA, thus leading to the development defects of pre-ZGA embryos. However, the difference is that studies on pigs, sheep, mice, zebrafish, and other species have shown that the development of embryos is affected by m6A modification, while the mechanism of the effect on cattle has not been reported [68,69,70,71]. CENPF, ZSCAN4, GLIS1, TEAD4, and CDX2 all inhibited the development of early ZGA to varying degrees. For example, CENPF-specific knockdown resulted in mRNA and protein silencing during pre-implantation development, disrupting the morphology of blastomere and inhibiting the development of 8 cells after implantation. ZSCAN4 knockdown affected PIWIL2 mRNA level, which may affect the normal function of transposons and lead to a decrease in the number of 16-cell stage embryos. Down-regulation of GLIS1 affected the expression levels of PDHA1 and HSPA8, thus inhibiting the embryonic development from the 16–32 cell stage. TEAD and CDX2 should interact with each other to ensure the stable expression of the TE gene; otherwise, the transcriptional regulation of pluripotency-related genes in bovine blastula TE and ICM cell lines will be affected, leading to the failure of embryonic development. USF1, on the other hand, alters the developmental capacity of oocytes by affecting the promoter-binding element E-box. In addition, the time of the first division after embryo fertilization is also closely related to the normal development of the embryo. If the time interval of the first division after embryo fertilization is too long, the igf-1 ligand may be reduced or even absent, and the mRNA abundance may be changed in response to the unfavorable growth environment, thus adversely affecting the development of the embryo [72,73]. The role of the fallopian tube in the development of the early embryo should not be ignored, and the mechanism of embryo-fallopian tube interaction also affects changes in transcription levels [74,75]. We can see that there is up-regulation and down-regulation of gene expression before embryo implantation. During this period, no matter how large or small the gene is, once the expression disorder occurs, the embryo development will encounter problems. Therefore, the research on important genes related to genes before embryo implantation is of great significance and also poses great challenges. However, as technology advances and we learn more about genes and the mechanisms by which they work, the more comprehensive the study will be.

## Figures and Tables

**Figure 2 ijms-24-16019-f002:**
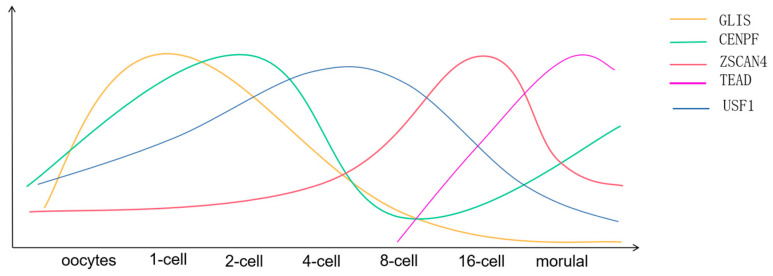
Description of expression patterns of CENPF, ZSCAN4, TEAD4, GLIS1, and USF1mRNA. The horizontal coordinate represents the stage of embryonic development; The ordinate only represents the rise and fall and has no actual value.

**Figure 3 ijms-24-16019-f003:**
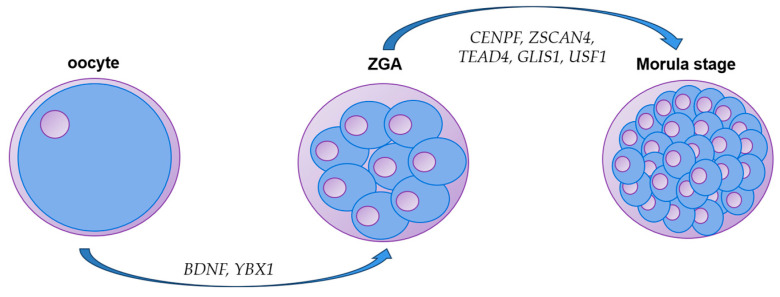
The process of the action of each gene is summarized. BDNF and YBX1 played an important role before ZGA. CENPF ZSCAN4, TEAD4 GLIS1, and USF1 affect the growth of the late ZGA.

## Data Availability

Not applicable.

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
