# Peer review of "The Role of BDNF, YBX1, CENPF, ZSCAN4, TEAD4, GLIS1 and USF1 in the Activation of the Embryonic Genome in Bovine Embryos"

_ijms, 2023, doi:10.3390/ijms242216019_

Round 1

Reviewer 1 Report

Comments and Suggestions for Authors

 Developmental genetic programs of bovine early embryogenesis

‘’Many maternal genes are synthesized and accumulated 57 in the oocyte and play a key role in early embryonic development’’

Very generally written. Could the most important genes be mentioned

Line 70-73: Why were these genes selected? What do the authors want to convey? This should be written more clearly.

Line 96-99: The name of the Authors? Or the number?

Line 112-117: Is that your interpretation? If yes, it needs to be written more clearly. If cited, the authors' names must be written.

Line 157-159: The origin of the protein (CENPF)?

Line 208-210: The topic is embryo. Sperm diploidosis is a different topic. Therefore the sentence must be deleted. Because the topic is much more complicated and needs clarification.

Line 221-222: In which animals? How TEAD4 causes embryo mortality (I would use the word embryo mortality instead of embryo death)

What is missing from the publication is an overview table. It can be the headings such as “BDNF =brain-derived neurotrophic factor” written in the table and the important genes added to them. The tasks or embryo development phase could be shown in another cell.

General: The work should be read again and the reasons and influence partly explained. The information such as ''Y-Box Binding Protein are reported to be enriched in oocytes'' is useful but the question of how or by which mechanism should be briefly addressed.

Conclusions:

The mechanisms and interactions should be discussed. 

Reviewer 2 Report

Comments and Suggestions for Authors

In this review work the authors aimed to report the results of recent findings on the timing and mechanisms of gene expression in bovine zygotes/embryos. The work results are a little heavy to read. The introduction needs to be improved. Numerous points need to be clarified and concerns to be addressed.

Introduction: This section could be expanded, and more specific references could be given to bovine embryonic development and not just to mammals in general. The authors did not mention any studies concerning the genetic program of embryonic development of the bovine. The aim of the study must be to better explain. In this way, it is not clear if this study concerns the genetic program of early embryogenesis development in cattle or mammals in general.

Line 38-39: it must be rewritten. There is a repetition of words, and it is not clear if ZGA regards zygotic/embryonic genome activation or zygotic gene activation.

Line 44: ZGA does not correspond to the first citation in the previous lines.

Line 49: This sentence must be rewritten. It is not clear the concept. What does “genes of germ cells (..) altered in mammalian gametogenesis and early embryonic development” mean?

Line 77: remove the “oocytes” extra.

Line 79-83: It must be rewritten. It is not clear.

Line 140 (Figure 1) moves the figure legend.

Lines 207-208: the sentence is not complete.

Line 210-212: the authors stated, “It can be speculated that the down-regulation of PIWIL2 expression will cause the dysfunction of other retrotransposons (including transcriptional transposons) in bovine embryos, and thus halt the early development of ZGA”. Why the down-regulation of PIWIL2 expression could determine the dysfunction of other retrotransposons? Add an explanation or reference/s supporting this evidence.

Sections 2,3,4: Since the work results are heavy to read in my opinion the authors could add a table that summarizes the genetic factors associated with impediments to oocyte maturation that have been identified in recent years.

Line 305 (Figure 2): add a reference.

Line 320: Figure?

Round 2

Reviewer 2 Report

Comments and Suggestions for Authors

The authors did great work to improve the manuscript. They addressed all the points of this round of revision. In my opinion,  this version of the manuscript fits the standard of the journal for publication.

Author Response

Thank the reviewers for the time and effort they put into reviewing the manuscript. Your suggestions are very important to improve the quality of the article.